# Radial Variation in Wood Anatomy of *Cercis glabra* and Its Application Potential: An Anatomy-Guided Approach to Sustainable Resource Utilization

**DOI:** 10.3390/plants14172769

**Published:** 2025-09-04

**Authors:** Pingping Guo, Xiping Zhao, Dongfang Wang, Yuying Zhang, Puxin Xie, Tifeng Zhao, Xinyi Zhao, Xinyi Lou

**Affiliations:** College of Horticulture and Plant Protection, Henan University of Science and Technology, Luoyang 471023, China; 9903545@haust.edu.cn (P.G.); 230320191164@stu.haust.edu.cn (D.W.); 240320191213@stu.haust.edu.cn (Y.Z.); 231419040220@stu.haust.edu.cn (P.X.); 231419040128@stu.haust.edu.cn (T.Z.); 241419040229@haust.edu.cn (X.Z.); 241419040216@haust.edu.cn (X.L.)

**Keywords:** *Cercis glabra*, radial wood anatomy variation, functional–structural trade-off, sustainable forestry, multipurpose utilization

## Abstract

This study systematically analyzes the microstructure and radial variation of *Cercis glabra* wood, revealing its adaptive strategies for arid environments. The results show that the wood consists of thick-walled fibers (63%) and vessels (17.7%), with a semi-ring-porous structure and 48.4% average cell wall percentage. Fiber proportion peaks early (4 years), ensuring mechanical support, while vessel adjustment occurs later (19 years), balancing water transport. Rays decline sharply in the first 9 years, stabilizing thereafter, reflecting a shift from growth to structural stability. The high fiber proportion and occasional tyloses enhance durability, making it suitable for high-quality pulp, furniture, and humid environments such as shipbuilding. A rotation period ≥ 20 years ensures stable properties. Genetic breeding could shorten the juvenile stage and optimize vessel distribution. Future research should integrate multi-omics and environmental data to deepen our understanding of its adaptation mechanisms. This study provides a basis for the utilization of *C. glabra* resources.

## 1. Introduction

*Cercis glabra*, also known as *Cercis gigantea* Cheng et Keng f. [1], is a deciduous *Fabaceae* tree endemic to China, mainly distributed in Hubei, Henan, and Shaanxi provinces. Because of its smooth leaves, *C. glabra* is internationally known as smooth redbud. Because of its tall plant, *C. glabra* is commonly used in China as giant redbud and also as Hubei redbud [2]. Recent ecological and cultural revitalization efforts have highlighted its multifunctional value in habitat restoration, landscape design, pharmaceutical applications, and biomass utilization, requiring comprehensive research to unlock its full potential.

*C. glabra* exhibits strong adaptability to nutrient-poor soils, with its developed root system and nitrogen fixation capacity effectively improving soil structure [3]. Studies [4] demonstrate its role as a pioneer species in rapidly colonizing degraded areas, controlling erosion, and enhancing soil fertility—key to subsequent vegetation succession. Under high greenhouse gas emission scenarios, projected climate shifts further expand its suitable habitat range [5]. These characteristics make *C. glabra* an ideal candidate for ecological restoration in low-fertility environments.

*C. glabra* has long been cultivated as an ornamental tree in traditional Chinese courtyards across Hubei and Anhui provinces. Its landscape value has gained renewed recognition in contemporary urban greening projects. Major municipal initiatives in cities such as Xi’an, Shijiazhuang, and Jingzhou now feature *C. glabra* prominently as a preferred street tree species [6], demonstrating its growing importance in modern urban forestry applications.

*C. glabra* is rich in bioactive compounds including isoflavones and terpenoids. Shu, et al. [7] demonstrated its extract’s significant radical scavenging capacity and anti-inflammatory properties in vitro. Subsequent studies have identified condensed tannins as key active components [8], highlighting their potential for antioxidant and anti-inflammatory applications. While current research remains at the laboratory stage, these findings suggest promising prospects for the development of natural anti-inflammatory therapeutics from this species.

*C. glabra* exhibits desirable characteristics for high-value applications, featuring a dense structure with white sapwood and dark-red heartwood. Its fine-grain texture, excellent toughness, and superior corrosion resistance make it particularly suitable for premium furniture and construction materials [9]. Despite these qualities, current utilization remains below 5%, mainly limited to fuelwood, reflecting a significant underutilization of its economic potential due to insufficient understanding of its anatomical properties.

Wood exhibits inherent structural anisotropy, with radial anatomical variation reflecting systematic changes in microstructure (vessels, fibers) along the stem radius. These variations reveal both physiological adaptations to environmental stresses (water, nutrients, light) and practical implications for forest resource utilization. Ecological studies demonstrate this plasticity—*Quercus robur* in water-rich Sava River sites develops larger fibers than in drier locations [10]. Such variations also affect carbon allocation strategies, potentially affecting forest carbon stock assessments [11]. Industrially, radial anatomical gradients determine critical quality parameters: juvenile wood near the pith typically exhibits shorter cells, thinner walls, and reduced diameters [12,13], resulting in inferior mechanical properties compared to mature wood [14,15]. These transitions serve as key indicators for optimizing wood processing and resource management.

Quantitative anatomical characterization is essential to optimize wood quality and commercial potential in high-value applications. This study establishes three key objectives: (1) to provide the first comprehensive anatomical profile of *C. glabra* wood according to the IAWA Hardwood Identification List [16], (2) to analyze patterns of the radial variation in critical structural parameters, (3) to identify the transition point from juvenile to mature wood through systematic anatomical assessment, and (4) to explore the possible link between anatomical characteristics and the commercial potential of *C. glabra* timber.

## 2. Results

### 2.1. Wood Anatomy of Cercis glabra

*C. glabra* wood exhibits a distinct bimodal cellular composition, consisting of thin-walled elements (vessels, rays, and axial parenchyma) interwoven with thick-walled fibers (Figure 1). Quantitative analysis revealed an average cell wall percentage (CP) of 48.4% (Table 1), reflecting its balanced structural organization.

Vessels constitute approximately 17.7% of the wood volume (Table 1), exhibiting a semi-ring-porous distribution with an average density of 103 vessels/mm^2^. The pores display circular to oval morphology, averaging 50 µm in diameter (radial: 55.2 µm, tangential: 46.6 µm), with earlywood vessels being nearly twice as wide as latewood vessels (Figure 1a). Vessel arrangements include solitary (45%), radial multiples of two–three cells (5%), and clustered formations (50%). Tyloses occasionally occlude the vessel lumen (Figure 1b), while vessel elements feature simple perforation plates (Figure 1c) and alternate intervessel pitting (Figure 1d).

The vessel elements are mainly embedded in thick-walled fibers (≈48%), with axial parenchyma constituting 7.61% of the tissue volume. In the cross-section, this parenchyma exhibits paratracheal-zonate distribution (Figure 1a), featuring cells with uniform radial and tangential diameters averaging 5 μm. Prismatic crystals occasionally occupy the axial parenchyma lumina (Figure 1e).

Rays constitute 16.7% of the wood volume, showing distinct structural characteristics across sections. Cross-sections reveal two–three cell-wide rays (Figure 1a), while tangential sections show both multiseriate and uniseriate types averaging 190 μm in height and 26.5 μm in width (Figure 1b). Radial sections demonstrate procumbent ray cells with occasional upright/square marginal cells (Figure 1f), measuring 48.3 μm in length and 9 μm in diameter. Irregularly shaped ray tracheids are occasionally present (Figure 1f).

### 2.2. Radial Variation in Wood Anatomy of Cercis glabra

#### 2.2.1. Cell Wall Percentage and Tissue Proportions

The cell wall percentage (CP) of *C. glabra* wood shows a distinct radial growth pattern, increasing progressively from pith to bark with a notable inflection point at 20 years (Figure 2). Linear piecewise regression reveals accelerated CP accumulation before this transition (R^2^ = 0.88), followed by a more gradual increase in mature wood (R^2^ = 0.92). Statistical analysis confirms a significant difference between juvenile (≈47% CP) and mature wood (≈50% CP), with a mean difference of 3% (*p* < 0.05, Table 2). This developmental shift mirrors the species’ transition from early structural establishment to later stabilization of wood properties.

CP showed significant correlations with wood tissue composition, showing a strong positive association with fiber proportion (FP) (r = 0.65, *p* < 0.01) while showing negative relationships with vessel proportion (VP) and ray proportion (RP) (Table 3). No significant correlation was observed between CP and axial parenchyma proportion (AP), suggesting its limited contribution to overall cell wall accumulation.

The fiber proportion (FP) exhibited a radial growth pattern similar to CP, with a rapid initial increase followed by stabilization, although its inflection point occurred much earlier (4 years vs. CP’s 20 years) (Figure 3a). Variance analysis confirmed significant FP differences between juvenile and mature wood (*p* < 0.001). Vessel proportion (VP) showed an inverse trend to CP, reaching its inflection point at 19 years with an initial decline followed by an increase (Figure 3b), showing a significant negative correlation with FP (*p* < 0.05) but no significant pre/post-inflection difference. Ray proportion (RP) decreased sharply in the first 9 years (juvenile phase, R^2^ = 0.05) before slowing (mature phase, R^2^ = 0.54) (Figure 3c), with significant maturity-stage differences (*p* = 0.009). Axial parenchyma (AP) showed no discernible radial pattern or inflection point (Figure 3d).

#### 2.2.2. Vessel Characteristics

Vessel density (VD) exhibited a characteristic radial pattern, initially decreasing before increasing from pith to bark, with an inflection point at 19 years (Figure 4a). Linear segmented regression demonstrated a good fit for both juvenile (R^2^ = 0.35) and mature wood (R^2^ = 0.69) stages. While the percentage of solitary vessels to all vessels (SVP) followed a similar radial trend to VD (Figure 4b), the percentage of clustered vessels to all vessels (CVP) showed an inverse pattern (Figure 4c), with both showing earlier inflection points (16 years). VD correlated positively with SVP (*p* = 0.01) and negatively with CVP (*p* = 0.05, Table 3). The percentage of multiple vessels to all vessels (MVP) increased radially (Figure 4d), while vessel lumen radial (VRD) and tangential diameter (VTD) decreased (Figure 4e,f), all showing marked variability without clear inflection points. ANOVA showed no significant differences in VRD or VTD between maturity stages (*p* = 0.794).

#### 2.2.3. Fiber Characteristics

The fiber diameter (FD) of *C. glabra* exhibited a distinct radial pattern, initially increasing before reaching a peak (10.29 μm at 18 years) and subsequently decreasing (Figure 5a). Linear piecewise regression demonstrated an excellent fit for both juvenile (R^2^ = 0.96) and mature wood (R^2^ = 0.82). Notably, the wall-to-lumen ratio (W/L) showed an inverse trend to FD, with a trough value (0.33) at 16 years (Figure 4b). Statistical analysis confirmed a significant negative correlation between FD and W/L (*p* < 0.05) and revealed higher values in mature wood for both parameters.

#### 2.2.4. Ray Characteristics

Radial variation in wood ray characteristics shows distinct developmental patterns (Figure 6). Both ray height (RH) and width (RW) show an initial decline followed by an increase with cambial age (CA), although their inflection points differ significantly, occurring at 10 and 18 years of age, respectively (Figure 6a,b). Ray parenchyma cells show a progressive enlargement in lumen diameter (RD) and length (RL) from pith to bark, with RL displaying a clearer developmental transition at 17 years (R^2^ = 0.06 pre-inflection vs. 0.40 post-inflection). RD maintains relatively consistent growth without distinct inflection points (Figure 6c).

#### 2.2.5. Axial Parenchyma Cell Diameter (AD)

AD exhibits a gradual decreasing trend from pith to bark with increasing CA, though the linear regression shows limited fitting accuracy (R^2^ = 0.27) without a distinct inflection point (Figure 7).

## 3. Discussion

### 3.1. Ecological Adaptability and Functional Correlation

The anatomical features of *C. glabra* wood, characterized by a 48.4% CP and semi-ring-porous vessel distribution, demonstrate its dual adaptation for mechanical strength and hydraulic efficiency. Vessels account for 17.7% of wood volume, mainly clustered (50%), while thick-walled fibers (FP: 48%) dominate, reflecting an arid-adaptive strategy: vessel clustering enhances hydraulic conductivity [17,18], and dense fibers provide structural resilience to drought stress [19]. To verify the arid-adaptive strategy, practical conducting efficiency and safety data also need to be detected for *C. glabra* trees in the future. Simple perforations and alternate intervessel pitting (Figure 1c,d) further streamline water transport [20,21], with occasional tyloses (Figure 1b) potentially acting as a pathogen defense mechanism [22].

The band-like distribution of axial parenchyma (Figure 1a) in *C. glabra* wood is similar to that of other *Fabaceae* wood [23]. Rich axial parenchyma and crystal deposition (Figure 1e,f) indicate potential roles in metabolite storage and ion homeostasis [24,25]. The heterocellular ray structure (16.7% by volume), combined with procumbent and upright cells (Figure 1f), enhances both radial transport efficiency [26] and crack resistance through structural heterogeneity [27]. Notably, the gradual earlywood–latewood vessel diameter transition (2:1 ratio) differs from typical diffuse/ring-porous woods, suggesting that this “semi-ring-porous” trait represents an adaptive compromise for seasonal drought and thermal variability [28,29]. In spring, rapid water delivery is required to meet the needs of new leaf expansion, while the small latewood vessel enhances structural stability to cope with summer drought [30].

### 3.2. Mechanism of Wood Maturity Revealed by Radial Variation

The radial variation in CP exhibits distinct segmentation, with a significant transition at year 20 (juvenile wood CP ≈ 47% vs. mature wood CP ≈ 50%, *p* < 0.05). While this pattern resembles *Sorbus alnifolia*’s maturation [31], *C. glabra*’s later transition (vs. 11 years in *S. alnifolia*) suggests delayed wood stabilization. The mechanical enhancement primarily stems from fiber thickening (CP-FP r > 0.8) rather than vessel adaptation (CP-VP negative correlation), reflecting a unique structural prioritization. The evolution of FP and CP is obviously the same. Wood fiber is the cell type, with a thick cell wall and a large W/L in wood. The increase in its proportion directly leads to an increase in the total amount of solid matter (cell wall) in wood.

Fiber morphology exhibits critical radial dynamics: FP inflection points at year 4 (preceding CP by 16 years) and post-15 W/L elevation (*p* < 0.03) demonstrate prioritized fiber wall thickening for early mechanical establishment [19]. This “fiber-first” strategy contrasts with delayed vessel optimization (VD/SVP/CVP inflection at 16–19 years), creating a developmental trade-off between structural reinforcement and hydraulic efficiency (Table 3).

*C. glabra*’s variation trend in the VP from pith to bark first decreases and then increases, which is consistent with the radial variation trend of the VP in *Paliurus hemsleyanus*, which is also semi-ring-porous wood [32]. This trend reflects similar ecological adaptation strategies between the two: the gradient of vessel size within semi-ring-porous wood rings is gentle, mechanical support is prioritized in the early stages (FP reaches its peak in the early stages), and vessel development is limited; the mature stage shifts towards hydraulic optimization, improving water delivery efficiency by increasing VP, achieving a dynamic balance between hydraulic safety and effectiveness [33].

From pith to bark, the RP of *C. glabra* shows an overall downward trend, similar to that of *Eucalyptus urophylla* [34]. However, the RP of *Tectona grandis* [35], *Larix gmelinii* [36], *Betula platyphylla* [37] remained essentially unchanged. Liu’s study [38] found that the RP of *Catalpa Bungei* clones showed a slight increase trend with an increase in CA. These results indicate that the variation in RP from pith to bark depends on the species. The dramatic early reduction in RP (30% decline within 9 years) followed by a gradual decrease (R^2^ = 0.54), coupled with divergent RH/RW patterns across transition points, suggests an ecological strategy shift: juvenile-phase high RP facilitates rapid radial expansion, while mature-phase RP reduction conserves metabolic resources [39]. Stable AP along with steadily decreasing AD requires continuous hydraulic optimization, maintaining safety margins without developmental inflection [40].

The current radial variation model does not account for interannual environmental variability (e.g., precipitation, temperature). Future studies should incorporate multi-site sampling with microclimate monitoring to develop robust anatomical–environmental response models.

### 3.3. Application of Wood Properties and Genetic Improvement Potential

The anatomical characteristics of *C. glabra* wood directly determine its functional properties and commercial potential. Mature wood exhibits superior mechanical performance due to its high FP (about 48%) and CP (50%), with secondary wall thickening (W/L peak: 0.33) significantly enhancing bending strength for structural applications [41]. Its 50% clustered vessel distribution, combined with semi-ring-porous characteristics, optimizes both hydraulic efficiency and dimensional stability through dense fiber arrangement, ideal for joinery requiring deformation resistance. Wood’s natural durability stems from high CP and occasional tyloses that inhibit microbial colonization [22], extending service life in marine and outdoor applications.

The processing characteristics of *C. glabra* wood are jointly determined by vessel and fiber morphology. Mature wood’s low W/L facilitates clean slicing and carving operations, while increased clustered vessel proportion minimizes cell wall tearing during machining. However, large vessels (>60 μm) require controlled chemical penetration for optimal papermaking [42]. The uniform texture resulting from its semi-ring-porous structure further enhances suitability for the manufacture of premium furniture.

Our findings suggest that the juvenile–mature wood transition occurs around the 20th growth ring, indicating significant juvenile wood content in potential yields. Strategic genetic improvement should prioritize the following: (1) implementation of ≥20-year rotation cycles to ensure material quality stability, and (2) accelerated maturation through targeted breeding of *C. glabra*.

It should be noted that this study only considered the anatomical characteristics of *C. glabra* wood, with the analysis based on anatomical characteristics being rather arbitrary, and it can only be treated as an estimate. Since this does not constitute a comprehensive study of wood properties that determine wood for utilization purposes, a more complete analysis of the chemical components, physical and mechanical properties should be carried out in the future.

## 4. Materials and Methods

### 4.1. Experimental Materials

This study utilized six *C. glabra* trees collected from Wumasi Forest Farm (111°58′ E, 33°43′ N) in Henan Province, China. This transitional ecotone between warm-temperate and northern subtropical zones features a continental monsoon climate (14 °C mean annual temperature, 700 mm precipitation, 236-day frost-free period) at 1500 m average elevation, supporting 98.5% forest cover with high biodiversity. Increment cores were extracted at breast height (1.3 m) from the southern side of each tree for age determination through ring counting (Table 4).

### 4.2. Anatomic Measurement

The increment cores were immersed in a 1:1 glycerol–alcohol solution for softening. The softened cores were sectioned annually from pith to bark, embedded in paraffin, and sliced into 12 μm sections using a Leica RM2235 slicing machine (Leica Microsystems, Wetzlar, Germany). Cross-sections were doubly stained with safranin and fast green to enhance differentiation between axial parenchyma and fibers, while tangential and radial sections received safranin-only staining for anatomical visualization.

The anatomical analysis was performed using the Mshot MD6.4 digital imaging system (Micro-shot Technology, Guangzhou, China) for microscopic observation and documentation, followed by quantitative measurements with the TDY-5.2 Wood Anatomy Measurement System (Beijing Tian Di Yu Technology Co., Ltd., Beijing, China). According to the method described by Yu et al. [43], the images obtained were analyzed by threshold, binary transform, grain filter, grain expansion and erosion, lacunae calking, and morphological feature calculation. Tangential sections were used to quantify ray height and width (≥3 rays per annum ring), while radial sections assessed ray parenchyma cell size (≥30 cells per annum ring). Transverse sections provided data on tissue proportions (cell wall, fiber, vessel, ray, and axial parenchyma ratios), fiber lumen diameter and wall-to-lumen ratio, vessel characteristics (radial/chordal diameters, density, solitary/grouped percentages), and axial parenchyma lumen diameter (≥60 cells per parameter per ring) to ensure statistical robustness.

### 4.3. Statistical Analyses

Statistical analyses were performed using SPSS (v24.0, IBM Corp., Armonk, NY, USA) to compute descriptive statistics (mean, range, SD, CV) for anatomical parameters. Correlation analysis assessed inter-trait relationships, while segmented linear regression identified juvenile-to-mature transition points by growth ring [31,44]. ANOVA with Duncan’s post hoc test evaluated maturity-stage differences in wood anatomical features.

## 5. Conclusions

The anatomical architecture and developmental trajectories of *Cercis glabra* wood collectively demonstrate a “drought adaptation-property optimization” strategy: juvenile-phase prioritization of fiber wall thickening for structural reinforcement transitions to mature-phase vessel refinement for hydraulic efficiency. These findings advance xylem adaptation theory while supporting practical applications. Wood’s high fiber proportion and dimensional stability make it possible to manufacture high-quality paper and furniture.

It should be noted that this study only considered the anatomical characteristics of *C. glabra* wood, and the analysis based on anatomical characteristics is rather arbitrary and can only be treated as an estimate. Since this does not constitute a comprehensive study of wood properties that determine wood for utilization purposes, a more complete analysis of the chemical components, physical and mechanical properties should be carried out in the future. In addition, future investigations should also employ multi-omics approaches to decode the molecular drivers of radial variation, enabling targeted breeding to accelerate maturation and enhance the species’ economic potential.

## Figures and Tables

**Figure 1 plants-14-02769-f001:**
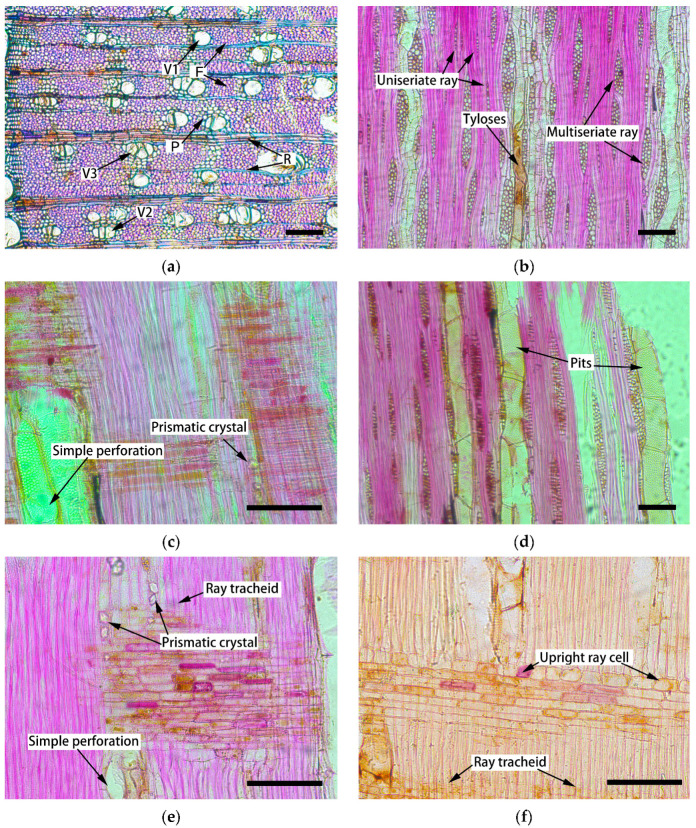
Sections of xylem in *Cercis glabra*: (**a**) cross-section, wood was semi-ring-porous with vessels being solitary (V1), radial multiples (V2), and clustered (V3), axial parenchyma (P) in paratracheal-zonate, ray (R), fiber (F); (**b**) tangential section, tyloses in vessels, multiseriate and uniseriate rays; (**c**) radial section, simple perforation of vessel elements; (**d**) tangential section, simple and alternates pits on the vessel wall; (**e**) radial section, prismatic crystals in the axial parenchyma cells, ray tracheids, simple perforation of vessel elements; (**f**) radial section, prismatic crystals in the axial parenchyma cells, upright ray cells. The scale bar represents 100 µm.

**Figure 2 plants-14-02769-f002:**
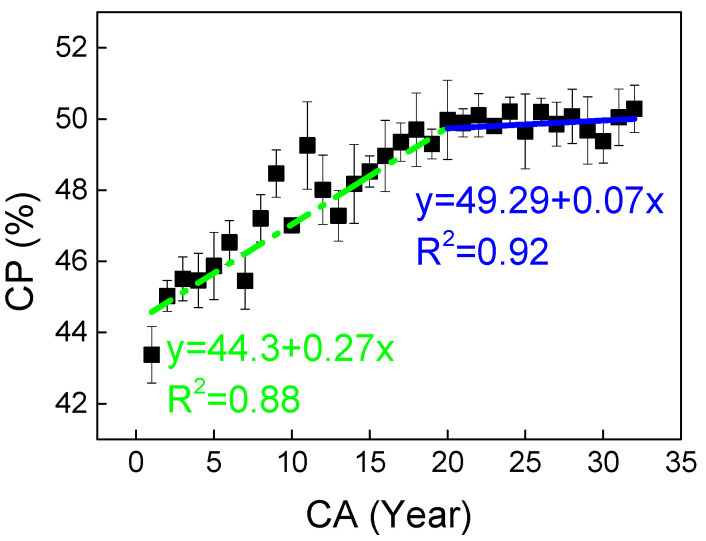
Observed (mean) and two segment-fitted cell wall percentage (CP) trends vs. cambial age (CA) in *Cercis glabra* wood. Bars show standard error. Green, dash-dot lines represent the first segment (juvenile wood). Blue, solid lines represent the second segment (mature wood). Confidence interval: 95%.

**Figure 3 plants-14-02769-f003:**
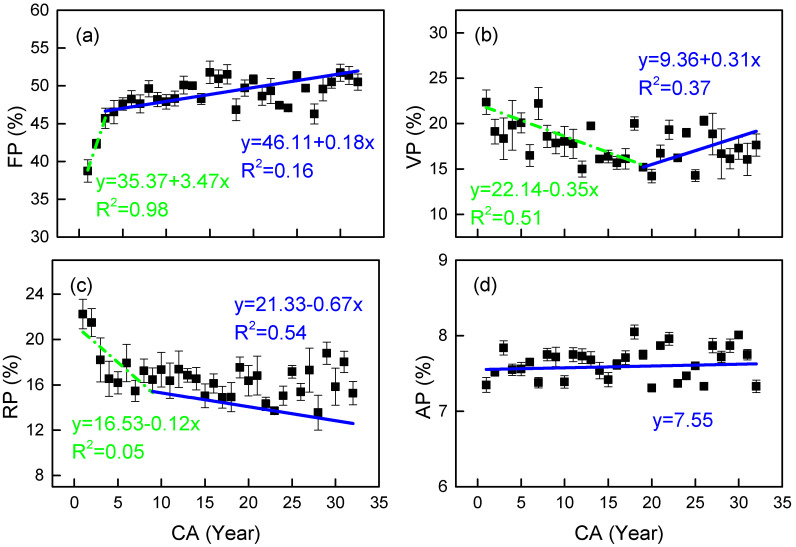
Observed (mean) and two segment-fitted tissue proportion trends vs. cambial age (CA) in *Cercis glabra* wood. Bars show standard error. Green, dash-dot lines represent the first segment (juvenile wood). Blue, solid lines represent the second segment (mature wood). (**a**) FP represents the fiber proportion; (**b**) VP represents the vessel proportion; (**c**) RP represents the ray proportion; (**d**) AP is the axial parenchyma tissue proportion. Confidence interval: 95%.

**Figure 4 plants-14-02769-f004:**
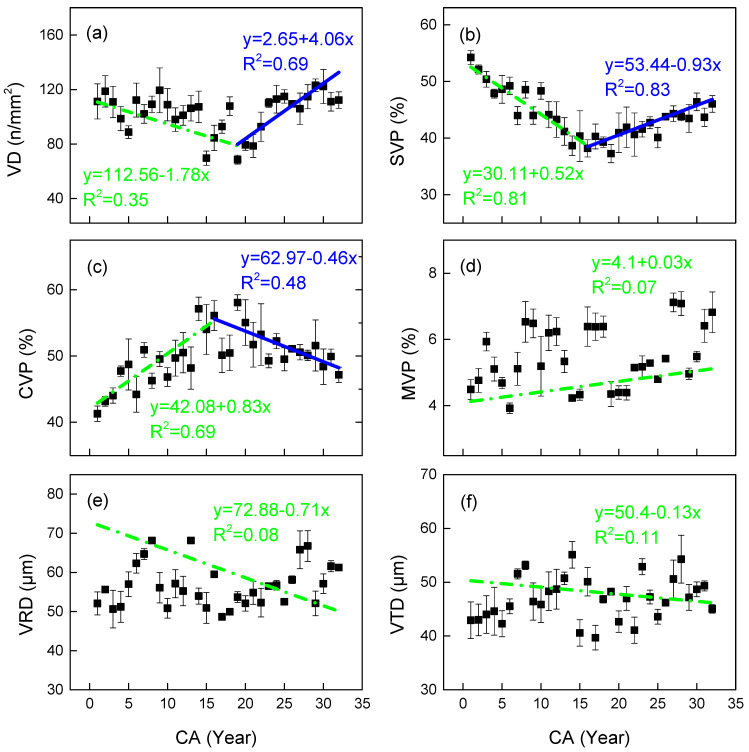
Observed (mean) and two segment-fitted vessel characteristics trends vs. cambial age (CA) in *Cercis glabra* wood. Bars show standard error. Green, dash-dot lines represent the first segment (juvenile wood). Blue, solid lines represent the second segment (mature wood). (**a**) VD represents the vessel density; (**b**) SVP represents the percentage of solitary vessels to all vessels; (**c**) CVP represents the percentage of clustered vessels to all vessels; (**d**) MVP represents the percentage of multiple vessels to all vessels; (**e**) VRD represents the vessel lumen radial diameter; (**f**) VTD is the vessel lumen tangential diameter. Confidence interval: 95%.

**Figure 5 plants-14-02769-f005:**
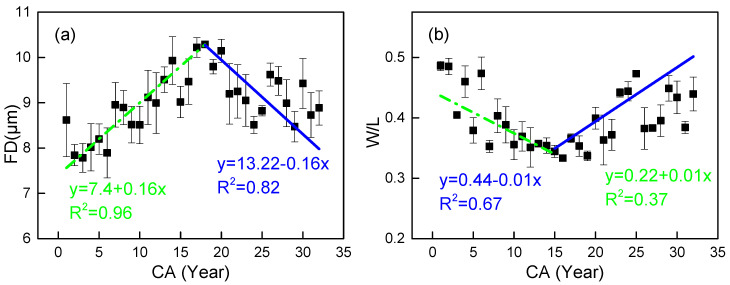
Observed (mean) and two segment-fitted FD and W/L trends vs. cambial age (CA) in *Cercis glabra* wood. Bars show standard error. Green, dash-dot lines represent the first segment (juvenile wood). Blue, solid lines represent the second segment (mature wood). (**a**) FD represents the fiber lumen diameter; (**b**) W/L is the ratio of fiber double wall thickness to fiber lumen diameter. Confidence interval: 95%.

**Figure 6 plants-14-02769-f006:**
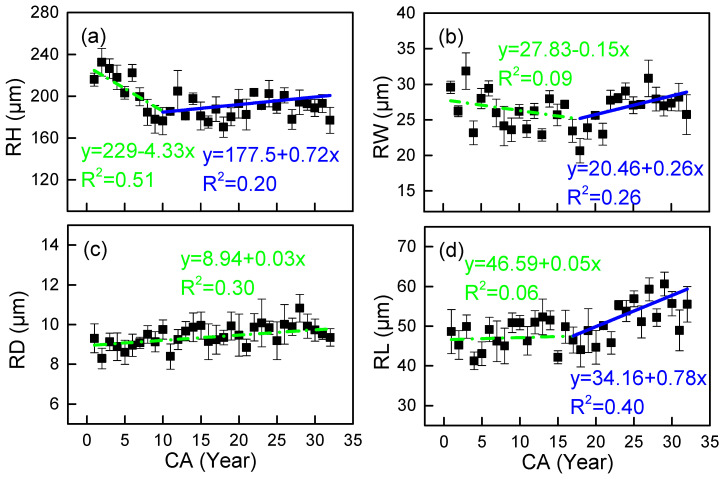
Observed (mean) and two segment-fitted vessel characteristics trends vs. cambial age (CA) in *Cercis glabra* wood. Bars show standard error. Green, dash-dot lines represent the first segment (juvenile wood). Blue, solid lines represent the second segment (mature wood). (**a**) RH represents the ray height; (**b**) RW represents the ray width; (**c**) RD represents the parenchyma cell lumen diameter; (**d**) RL is the parenchyma cell length. Confidence interval: 95%.

**Figure 7 plants-14-02769-f007:**
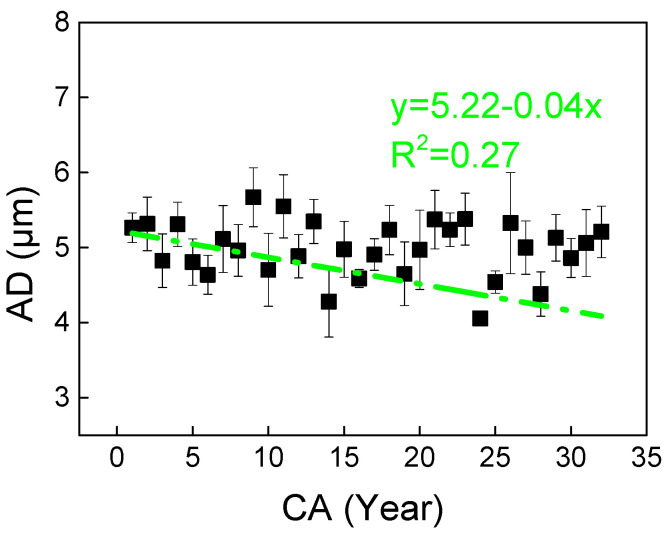
Observed (mean) and line-fitted axial parenchyma cell lumen diameter (AD) trends vs. cambial age (CA) in *Cercis glabra* wood. Bars show standard error. Green, dash-dot lines represent the linear fitting equation. Confidence interval: 95%.

**Table 1 plants-14-02769-t001:** Descriptive statistics of the anatomical characteristics in *Cercis glabra* wood.

Characteristics	Minimum	Maximum	Average	SD	CV (%)
CP (%)	41.16	53.67	48.36	2.61	5.40
VP (%)	11.46	30.37	17.74	3.68	20.74
FP (%)	32.26	58.72	47.96	4.29	8.94
AP (%)	6.00	9.69	7.61	0.73	9.59
RP (%)	9.54	26.84	16.67	3.46	20.76
VD	49.11	176.85	103.2	25.95	25.15
SVP (%)	25.0	59.18	44.60	6.48	14.53
CVP (%)	34.01	66.67	49.89	6.75	13.53
MVP (%)	2.70	8.70	5.35	1.24	23.18
VRD (µm)	30.41	69.65	55.20	9.05	16.39
VTD (µm)	28.40	60.51	46.63	6.95	14.90
FD (n/mm^2^)	5.71	11.52	8.98	1.22	13.59
W/L	0.22	0.54	0.38	0.07	18.42
RH (µm)	114.32	281.85	189.55	37.40	19.73
RW (µm)	15.35	38.57	26.46	4.37	16.52
RD (µm)	5.92	13.17	9.31	1.47	15.79
RL (µm)	23.47	67.66	48.34	8.93	18.47
AD (µm)	2.87	7.20	5.02	0.86	17.13

Note: CP, Cell Wall Percentage; FP, Fiber Proportion; VP, Vessel Proportion; RP, Ray Proportion; AP, Axial Parenchyma Proportion; VD, Vessel Density; SVP, Percentage of Solitary Vessels to All Vessels; CVP, Percentage of Clustered Vessels to All Vessels; MVP, Percentage of Multiple Vessels to All Vessels; VRD, Vessel Lumen Radial Diameter; VTD, Vessel Lumen Tangential Diameter; FD, Fiber Lumen Diameter; W/L, Ratio of Fiber Wall Thickness to Fiber Lumen Diameter; RH, Ray Height; RW, Ray Width; RD, Ray Cell Lumen Diameter; RL, Ray Cell Length; AD, Axial Parenchyma Cell Lumen Diameter. SD, Standard Deviation; CV, Coefficient of Variation.

**Table 2 plants-14-02769-t002:** Anatomical differences in the juvenile and mature wood in *Cercis glabra* (mean ± Std).

Characteristics	Juvenile Wood	Mature Wood	*p*-Value
CP (%)	47.29 ± 1.80	49.93 ± 0.26	0.000
VP (%)	18.32 ± 2.14	16.99 ± 1.87	0.076
FP (%)	42.26 ± 3.48	49.32 ± 2.83	0.000
AP (%)	—	—	—
RP (%)	17.98 ± 2.37	16.11 ± 1.38	0.009
VD	102.59 ± 12.53	103.97 ± 17.19	0.794
SVP (%)	45.80 ± 4.80	42.25 ± 2.47	0.013
CVP (%)	48.14 ± 4.16	51.44 ± 2.81	0.013
MVP (%)	—	—	—
VRD (µm)	—	—	—
VTD (µm)	—	—	—
FD (n/mm^2^)	8.70 ± 0.65	9.31 ± 0.59	0.010
W/L	0.43 ± 0.05	0.49 ± 0.04	0.028
RH (µm)	208.90 ± 18.86	188.07 ± 9.87	0.000
RW (µm)	26.36 ± 2.56	26.40 ± 2.58	0.961
RD (µm)	—	—	—
RL (µm)	47.70 ± 3.56	51.82 ± 5.16	0.013
AD (µm)	—	—	—

Note: CP, Cell Wall Percentage; FP, Fiber Proportion; VP, Vessel Proportion; RP, Ray Proportion; AP, Axial Parenchyma Proportion; VD, Vessel Density; SVP, Percentage of Solitary Vessels to All Vessels; CVP, Percentage of Clustered Vessels to All Vessels; MVP, Percentage of Multiple Vessels to All Vessels; VRD, Vessel Lumen Radial Diameter; VTD, Vessel Lumen Tangential Diameter; FD, Fiber Lumen Diameter; W/L, Ratio of Fiber Wall Thickness to Fiber Lumen Diameter; RH, Ray Height; RW, Ray Width; RD, Ray Cell Lumen Diameter; RL, Ray Cell Length; AD, Axial Parenchyma Cell Lumen Diameter. —, no significant differences in VRD or VTD between maturity stages.

**Table 3 plants-14-02769-t003:** Correlation among the anatomical characteristics of *Cercis glabra* wood.

Characteristics	CP	FP	VP	AP	RP	VD	SVP	CVP	MVP	VRD	VTD	FD	W/L	RH	RW	RL	RD
FP	0.65 **																
VP	−0.5 **	−0.66 *															
AP	0.26	0.22	−0.12														
RP	−0.6 **	−0.4 **	0.15	−0.04													
VD	−0.11	−0.15	0.24	0.08	0.21												
SVP	−0.7 **	−0.5 **	0.51 **	−0.17	0.59 **	0.46 **											
CVP	0.61 **	0.53 **	−0.5 *	0.07	−0.5 *	−0.6 *	−0.9 **										
MVP	0.32	0.19	0.04	0.33	−0.29	0.30	−0.03	−0.09									
VRD	0.03	0.004	0.18	0.004	−0.11	0.24	0.10	−0.12	0.39 *								
VTD	0.17	0.06	−0.06	0.12	−0.19	0.27	−0.21	0.21	0.30	0.64 **							
FD	0.57 **	0.29	−0.23	0.14	−0.44 *	−0.38 *	−0.7 **	0.64 **	0.15	−0.05	0.19						
W/L	−0.27	−0.28	0.11	−0.20	0.42 *	0.58 **	0.59 **	−0.6 **	−0.21	−0.05	−0.22	−0.6 *					
RH	−0.6 **	−0.4 **	0.26	−0.18	0.45 **	0.25	0.61 **	−0.5 *	−0.38 *	−0.14	−0.27	−0.6 **	0.57 **				
RW	−0.14	−0.12	0.02	−0.15	0.21	0.30	0.36 *	−0.20	−0.06	0.12	0.04	−0.40	0.34	0.48 *			
RL	0.41 *	0.19	−0.27	0.12	0.02	0.52 *	−0.12	0.02	0.21	0.23	0.37 *	0.02	0.24	−0.23	0.40 *		
RD	0.49 **	0.36 *	−0.18	0.001	−0.42 *	0.06	−0.38 *	0.38 *	0.14	0.16	0.38 *	0.36 *	−0.17	−0.32	0.23	0.43 *	
AD	−0.10	−0.14	0.35	−0.1	0.13	0.03	0.18	−0.29	0.11	−0.03	−0.19	−0.02	0.04	−0.07	−0.43 *	−0.19	−4.35

Note: CP, Cell Wall Percentage; FP, Fiber Proportion; VP, Vessel Proportion; RP, Ray Proportion; AP, Axial Parenchyma Proportion; VD, Vessel Density; SVP, Percentage of Solitary Vessels to All Vessels; CVP, Percentage of Clustered Vessels to All Vessels; MVP, Percentage of Multiple Vessels to All Vessels; VRD, Vessel Lumen Radial Diameter; VTD, Vessel Lumen Tangential Diameter; FD, Fiber Lumen Diameter; W/L, Ratio of Fiber Wall Thickness to Fiber Lumen Diameter; RH, Ray Height; RW, Ray Width; RD, Ray Cell Lumen Diameter; RL, Ray Cell Length; AD, Axial Parenchyma Cell Lumen Diameter. Asterisks indicate the statistical significance of the correlations: * *p* < 0.05, ** *p* < 0.01.

**Table 4 plants-14-02769-t004:** Basic condition of sampling trees (mean ± Std).

Tree Age	Diameter at Breast Height (cm)	Height (m)	Height of the Lowest Branch (m)
23 ± 4.7	26.4 ± 7.0	22 ± 3.3	15.2 ± 3.6

## Data Availability

The original contributions presented in this study are included in the article. Further inquiries can be directed to the corresponding authors.

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
