# Peer review of "Radial Variation in Wood Anatomy of Cercis glabra and Its Application Potential: An Anatomy-Guided Approach to Sustainable Resource Utilization"

_plants, 2025, doi:10.3390/plants14172769_

Round 1
Reviewer 1 Report
Comments and Suggestions for Authors
This review critically evaluates the anatomical analysis of wood presented in the manuscript, highlighting concerns regarding methodological clarity, data accuracy, and interpretative overreach.
Is there any common English name for Cercis glabra? You used only Latin names throughout the text. You should use some synonyms like Redbud.
Were the proportions of individual cells measured in selected annual growth rings or along the entire length of the core sample? Similarly, regarding the dimensions of the cells—were they measured within individual increments or along the whole length of the core? What criterion was used to select those increments for analysis within individual growth rings?
Figure 1: Some of the figure labels are not visible against the background of the cells; please consider changing, for example, the font colour—especially the labels in Figure 1a.
Lines 98-99: The proportion of wood parenchyma is only 2%, but the analysis of image 1a unfortunately gives the misleading impression that there is significantly (two to three times) more parenchyma. Near almost every vessel, there is a cluster of parenchyma cells in which several or even a dozen cells can be counted. The left part of photograph 1a also gives the impression that a layer of terminal parenchyma is beginning near the end of the annual ring, which further contributes to the perception of a higher amount of parenchyma. Is the percentage determined as 2% accurate and free from error?
Table 1: You start the description below with CA - cambial age, which is unavailable in this table.
Table 2: You wrote that CP for mature wood is 19.93, while for juvenile wood, 47.29, which looks like a mistake, because on figure 2, which is above, we can observe a different trend. In the same table, errors affect the rest of the values. In Table 1, it is stated that the average FP is 63.44, but in juvenile wood it is 18.32, and in mature wood 16.99, which does not yield the previously mentioned result when using a weighted average. Please verify the accuracy of the results presented in the tables.
Figure 3: How do you explain the changes in the wood presented in photographs 3b and 3c? Why does the vessel proportion decrease until year 20 and then increase, reaching a value nearly the same as in juvenile wood? Why does the proportion of wood rays decrease by about 10%? Usually, as the stem diameter and circumference increase, the number of wood rays increases as well, to effectively transport substances.
The discussion section is full of speculation. The authors claim that due to specific anatomical parameters, the wood will have a positive effect on bending strength (p. 272). However, we cannot know how it truly performs without actually testing and determining this strength. Similarly, earlier (pp. 207–210), the authors' analyses cannot answer how the tree will behave during drought or how efficiently it transports water. In the reviewer’s opinion, the obtained results do not differ significantly from those of other wood species originating from monsoon forests. This is a typical wood structure, and due to the tree’s aesthetic qualities (beautiful flowers), it is rarely felled and used in industry. At the end of the discussion section, the authors argue that the fibre arrangement and cell wall proportions will contribute to good dimensional stability and expand the potential applications of this wood (pp. 272–277). This is an overinterpretation, and such a claim should be verified through proper testing and research beforehand.
What was the age of the analysed trees? There is no clear information on this matter.
Do you consider macerating material and measuring the length of fibres, vessels and other cells?
Lines 327-329: A much more useful parameter for wood intended for paper production would be the cellulose content and secondary compounds, not necessarily the proportion of fibres.
Author Response
Dear Reviewer:
Thank you for your comments on our paper (Plants -3793197). We greatly admire your profound professional knowledge and rigorous scientific attitude. What is more valuable is that you not only pointed out the problems in our manuscript, but also provided valuable suggestions for modification. The main corrections with red text in the paper and responses to your comments are as follows:
- Is there any common English name for Cercis glabra? You used only Latin names throughout the text. You should use some synonyms like Redbud.
Response to (1): In the field of botany, the Latin scientific name is an official international name, while the English common name is less unified. Because of its smooth leaves, C. glabra is internationally known as smooth redbud. Because of its tall plant, C. glabra is commonly used in China as giant redbud and also as Hubei redbud. Following your suggestion, we added the English name in the Introduction.
- Were the proportions of individual cells measured in selected annual growth rings or along the entire length of the core sample? Similarly, regarding the dimensions of the cells—were they measured within individual increments or along the whole length of the core? What criterion was used to select those increments for analysis within individual growth rings?
Response to (2): The proportion of various cell types and cell size were measured in each growth ring. According to the method described by Yu et al. (2009), tangential sections were used to quantify ray height and width (≥3 rays per annum ring), while radial sections assessed parenchyma cell size (≥30 cells per annum ring). Transverse sections provided data on tissue proportions (cell wall, fiber, vessel, ray, and axial parenchyma ratios), fiber lumen diameter and wall-to-lumen ratio, vessel characteristics (radial/chordal diameters, density, solitary/group percentages), and axial parenchyma lumen diameter (≥60 cells per parameter per ring) to ensure statistical robustness. We are sorry for our less rigorous writing. We have corrected the word. Sorry, we missed this reference. We have supplemented this reference in the Materials and Methods section.
- Figure 1: Some of the figure labels are not visible against the background of the cells; please consider changing, for example, the font colour—especially the labels in Figure 1a.
Response to (3): According to your suggestion, we added the white background of the text in the figure to increase the readability of the font.
- Lines 98-99: The proportion of wood parenchyma is only 2%, but the analysis of image 1a unfortunately gives the misleading impression that there is significantly (two to three times) more parenchyma. Near almost every vessel, there is a cluster of parenchyma cells in which several or even a dozen cells can be counted. The left part of photograph 1a also gives the impression that a layer of terminal parenchyma is beginning near the end of the annual ring, which further contributes to the perception of a higher amount of parenchyma. Is the percentage determined as 2% accurate and free from error?
Response to (4): We especially admire your professional and academic skills. Indeed, the actual proportion of axial parenchyma may be higher. In the cross section (Figure 1a), except for axial parenchyma cells , the wall is thin , and the cross-sectional size and shape are similar to those of wood fibers. When we measure the proportion of axial parenchyma, we may misjudge it as parenchyma cells and thick walled cells adjacent to wood fibers, resulting in a low proportion of axial parenchyma. We carefully measured the proportion of axial parenchyma in each ring. The results showed that the proportion of axial parenchyma increased, while the proportion of fibrous tissue decreased. We corrected the measurement results of axial parenchyma proportion in the text, including the radial variation and correlation analysis with the measurement results.
- Table 1: You start the description below with CA - cambial age, which is unavailable in this table.
Response to (5): We are sorry for our less rigorous writing. Based on your kind suggestion, we have deleted "CA – cambial age".
- Table 2: You wrote that CP for mature wood is 19.93, while for juvenile wood, 47.29, which looks like a mistake, because on figure 2, which is above, we can observe a different trend. In the same table, errors affect the rest of the values. In Table 1, it is stated that the average FP is 63.44, but in juvenile wood it is 18.32, and in mature wood 16.99, which does not yield the previously mentioned result when using a weighted average. Please verify the accuracy of the results presented in the tables.
Response to (6): We are very sorry for our less rigorous writing. Based on your kind suggestion, we have corrected the data entry error in Table 2.
- Figure 3: How do you explain the changes in the wood presented in photographs 3b and 3c? Why does the vessel proportion decrease until year 20 and then increase, reaching a value nearly the same as in juvenile wood? Why does the proportion of wood rays decrease by about 10%? Usually, as the stem diameter and circumference increase, the number of wood rays increases as well, to effectively transport substances.
Response to (7): The change in VP in C. glabra from pith to bark is consistent with VP in Paliurus hemsleyanus, which are also semi-ring-porous wood (Fei 1994). This trend reflects similar ecological adaptation strategies between the two: the gradient of vessel size within semi-ring-porous wood rings is gentle, mechanical support is prioritized in the early stages (FP reaches its peak in the early stages), and vessel development is limited; the mature stage shifts towards hydraulic optimization, improving water delivery efficiency by increasing VP, achieving a dynamic balance between hydraulic safety and effectiveness (Liu et al., 2023).
As the diameter and circumference of the tree trunk increase, the total number of wood rays will also increase, but RP may not necessarily increase. RP variation from pith to bark depends on the species. Most tree species remain largely unchanged, such as Tectona grandis (Rahman et al., 2005), Larix gmelinii (Fonti et al., 2015), Betula platyphylla (Tang et al., 2018). Our research tree species, Bauhinia and Eucalyptus urophylla (Hu et al., 2008), are similar, with an overall decreasing trend in RP.
Based on your suggestion, we have added corresponding explanations to this part of the article.
- The discussion section is full of speculation. The authors claim that due to specific anatomical parameters, the wood will have a positive effect on bending strength (p. 272). However, we cannot know how it truly performs without actually testing and determining this strength. Similarly, earlier (pp. 207–210), the authors' analyses cannot answer how the tree will behave during drought or how efficiently it transports water. In the reviewer’s opinion, the obtained results do not differ significantly from those of other wood species originating from monsoon forests. This is a typical wood structure, and due to the tree’s aesthetic qualities (beautiful flowers), it is rarely felled and used in industry. At the end of the discussion section, the authors argue that the fibre arrangement and cell wall proportions will contribute to good dimensional stability and expand the potential applications of this wood (pp. 272–277). This is an overinterpretation, and such a claim should be verified through proper testing and research beforehand.
Response to (8): Fully mastering the anatomical structure of C. glabra wood is the basis for expanding the efficient utilization of wood, which is also the purpose of this paper. Many studies have shown that the anatomical structure is closely related to the physical and mechanical properties of wood. The discussion of this paper is based on other scholars' research results. Of course, as the reviewer pointed out, due to time and funding constraints, this paper does not focus on the water use effect and physical and mechanical properties of C. glabra, and the description section is based on anatomical structure speculation. In conclusion, we pointed out the limitations of the study. Next, we will conduct research in these aspects to verify our speculation.
- What was the age of the analysed trees? There is no clear information on this matter.
Response to (9): The age of the sample trees is shown in Table 3.
- Do you consider macerating material and measuring the length of fibres, vessels and other cells?
Response to (10): Cell length is also a key indicator of wood quality. For example, fiber length directly affects the tensile strength, flexural strength, and toughness of wood. In the paper industry, a fiber length of 0.9mm or more is required to meet the basic standards for papermaking raw materials. Short fibers may lead to insufficient paper strength, while long fibers may affect the uniformity of the resulting paper. However, there is a significant error in measuring the length of fibers or vessels in the longitudinal section. In the future, we will specifically use molecular isolation methods to measure the length of fibers, vessels, etc.
- Lines 327-329: A much more useful parameter for wood intended for paper production would be the cellulose content and secondary compounds, not necessarily the proportion of fibres.
Response to (11): You are right. FP is not the only factor in determining whether wood is suitable for papermaking. In addition, there are fiber length, fiber length/width, wall/lumen, cellulose content and secondary compounds you mentioned. Sorry, we write carelessly. We have corrected this conclusion.
Thank you again for your comments.
Reviewer 2 Report
Comments and Suggestions for Authors
General remarks:
This paper aims to describe the anatomy of Cercis glabra's wood and to assess how its anatomical features have evolved over the years along the growth rings, focusing on the transition from juvenile to mature wood.
The methodology used is accurate and current. The results are presented correctly. However, the organisation of the manuscript needs revision because "Materials and Methods" is placed before the conclusions, while "Results" comes directly after the Introduction.
Detailed remarks:
page 2, lines 72-73: Authors should describe the possible link between the anatomical characteristics and the commercial potential of a timber.
2.1. The description of the wood anatomy should follow the IAWA list, which is never quoted in this paper. On the contrary, they are a basic reference for the description of wood anatomy.
Page 3, lines 104-105: rays are apparently seriate on longitudinal sections (Figure 1, b and d).
Page 4, lines 127-133: some of the descriptions of radial variation seem pretty obvious. The evolution of the FP and CP is obviously the same: more fibres = more cell wall.
Page 8, lines 221-222: Is the distribution of axial parenchyma similar to that of other Fabaceae timber?
Page 9, lines 227-228: Is C. glabra a deciduous tree? Usually, this kind of anatomical adaptation is in connection with the new leaves’ production.
Page 11, line 291: This paragraph (Material and Methods) should be placed elsewhere. In any case, the authors should justify why sampling only six trees can yield reliable results, which could then support extending the findings to describe the wood of the entire species.
Page 11, lines 308-309: I could not find online any description of this tool. Could the author give a more complete description?
Author Response
Thank you for your comments on our paper (Plants -3793197). We greatly admire your profound professional knowledge and rigorous scientific attitude. What is more valuable is that you not only pointed out the problems in our manuscript, but also provided valuable suggestions for modification. The main corrections with red text in the paper and responses to your comments are as follows:
- The methodology used is accurate and current. The results are presented correctly. However, the organisation of the manuscript needs revision because "Materials and Methods" is placed before the conclusions, while "Results" comes directly after the Introduction.
Response to (1): Thank you for your recognition of our research methods and results. The organization of our manuscript was written in strict accordance with the template of the journal "Plants", in the order of 1. Introduction, 2. Results, 3. Discussion, 4. Materials and Methods, and 5. Conclusions.
- page 2, lines 72-73: Authors should describe the possible link between the anatomical characteristics and the commercial potential of a timber.
Response to (2): Based on your kind suggestion, we have added the content you suggested in the research section of the Introduction.
- The description of the wood anatomy should follow the IAWA list, which is never quoted in this paper. On the contrary, they are a basic reference for the description of wood anatomy.
Response to (3): We are sorry for omitting the reference “IAWA list”. Based on your kind suggestion, we have quoted the reference in the Introduction.
- Page 3, lines 104-105: rays are apparently seriate on longitudinal sections (Figure 1, b and d).
Response to (4): The arrangement of wood rays in wood is determined by its three-dimensional spatial structure and anatomical characteristics. Wood rays are radially and continuously arranged from the pith to the bark on the transverse section. The wood ray is cut into short lines, spindles or spots on the tangential section (one of the longitudinal sections) (depending on the height and width of the ray, Fig. 1, b and d). Wood rays are in continuous bands on the radial section (another type of longitudinal section), but may still be intermittent due to axial cell interpenetration and its own width changes (Fig. 1, c, e, and f).
- Page 4, lines 127-133: some of the descriptions of radial variation seem pretty obvious. The evolution of the FP and CP is obviously the same: more fibres = more cell wall.
Response to (5): You are right. Wood fiber is the cell type with a thick cell wall and a large W/L in wood. The increase in its proportion directly leads to an increase in the total amount of solid matter (cell wall) in wood. We added relevant content to the Discussion.
- T Page 8, lines 221-222: Is the distribution of axial parenchyma similar to that of other Fabaceae timber?
Response to (6): The zonal distribution of axial parenchyma in C. glabra wood is similar to that of other Fabaceae wood. We have supplemented the relevant supporting reference.
- Page 9, lines 227-228: Is glabra a deciduous tree? Usually, this kind of anatomical adaptation is in connection with the new leaves’ production.
Response to (7): You are right. C. glabra is a deciduous tree. Its semi-ring-porous characteristics are related to the generation of new leaves. In spring, rapid water delivery is required to meet the needs of new leaf expansion, while the small latewood vessel enhances structural stability to cope with summer drought. We have supplemented the relevant supporting reference.
- Page 11, line 291: This paragraph (Material and Methods) should be placed elsewhere. In any case, the authors should justify why sampling only six trees can yield reliable results, which could then support extending the findings to describe the wood of the entire species.
Response to (8): As mentioned earlier, the organizational order of our manuscript was written in strict accordance with the template of the journal "Plants". In the study of wood anatomy, the minimum number of samples should be determined comprehensively according to research objectives, species characteristics and methodological needs. Under single environmental conditions, the number of internationally samples is generally 3-5, and the statistical effect is ensured through variance analysis. Therefore, our study can obtain reliable results by sampling six trees.
- Page 11, lines 308-309: I could not find online any description of this tool. Could the author give a more complete description?
Response to (9): The images obtained were analyzed with TDY-5.2 Wood Anatomy Measurement System through the steps of threshold, binary transformation, grain filter, grain expansion and erosion, lacunae caving, and morphological feature calculation. Specific methods and steps are fully described in Yu (2009). Sorry, we missed this reference. We have supplemented this reference in the Materials and Methods section.
Thank you again for your comments.